# Beyond the Surface: A New Perspective on Dual-System Theories in Decision-Making

**DOI:** 10.3390/bs14111028

**Published:** 2024-11-01

**Authors:** Guy Hochman

**Affiliations:** Baruch Ivcher School of Psychology, Reichman University, Herzliya 4610101, Israel; ghochman@runi.ac.il; Tel.: +972-9-9602422

**Keywords:** intuitive, deliberative, processes, process-tracing techniques

## Abstract

The current paper provides a critical evaluation of the dual-system approach in cognitive psychology. This evaluation challenges traditional classifications that associate intuitive processes solely with noncompensatory models and deliberate processes with compensatory ones. Instead, it suggests a more nuanced framework where intuitive and deliberate characteristics coexist within both compensatory and noncompensatory processes. This refined understanding of dual-process models has significant implications for improving theoretical models of decision-making, providing a more comprehensive account of the cognitive mechanisms underlying human judgment and choice.

## 1. Introduction

Decision research scholars aim to reveal the cognitive processes underlying choice behavior. However, as mental processes are not observable, a variety of methodical developments have been proposed to overcome the obstacles to investigation. One major development is process modeling. Process models characterize the cognitive strategies employed by decision-makers while making judgments or choices [1,2]. Process-tracing techniques such as verbal protocol analysis, e.g., [3,4,5,6], information board, e.g., [7,8,9], and mouselab methodology, e.g., [10,11], and indirect measures of processes, e.g., [12,13,14,15], allow these models to trace the processes underling human decisions [16]. Direct and indirect measures are two approaches researchers use to understand the decision-making process. Direct measures involve straightforward methods where participants are explicitly asked about their thoughts, feelings, or actions (e.g., in surveys and interviews), or their responses are directly examined. Indirect measures, on the other hand, assess underlying processes without requiring participants to self-report. These methods often capture subtle, automatic reactions that participants may not be consciously aware of, such as physiological responses (like heart rate or pupil dilation) or behavioral indicators (like reaction times).

One major trend in decision research considers two families of cognitive processes: intuitive and deliberate [17,18,19,20,21]. In the current manuscript, I suggest two theoretical distinctions that might broaden the generality of this dual-system approach. The first distinction relates to certain decision situations in which conflicts between the two types of processes emerge [20]. Traditionally, the two-system theory explains such salient situations (e.g., the Müller-Lyer illusion [22] and the conjunction fallacy; for a full discussion, see [20,23]). However, situations where the dissociation between the two systems is less explicit are often neglected or ignored. I suggest that a consideration of these latent situations under the dual-process framework would account for some of the contradictory findings in the decision research literature (e.g., the findings about loss aversion [24] and other self-bias [25]). By revisiting such findings using a variety of physiological (autonomic) system measures, including pupil dilation, peripheral arterial tone, and heart rate, I propose the dissociation between behavior and autonomic responses as a marker of the different decision systems.

In addition, I suggest that the classical classification of cognitive processes into intuitive/deliberate and compensatory/noncompensatory, which are often considered analogous cf. [26,27], are, in fact, indicators of between-system and within-system processes. Briefly, there are two decision systems, each displaying compensatory and noncompensatory decision processes. Later, I will discuss the use of indirect measures of processes such as response time and choice proportion to highlight the robustness of this interpretation to account for several of the discrepancies in classical decision research (e.g., the theoretical hurdle faced by two-system approaches to successfully classify cognitive processes and heuristic tools into the two systems [26,28]; the findings that in some situations rational reasoning is the result of intuitive judgment [17,29,30,31,32,33,34,35,36]).

Considering the lack of a clear theoretical account and contradicting findings surrounding dual-system approaches [19,37], it is unsurprising that it attracts criticism [38,39]. Thus, a more refined and comprehensive theoretical account of the dual-system framework and a better understanding of the interplay between these two systems and their respective roles in reasoning are essential for advancing our comprehension of human cognition. In this paper, I present a critical review of dual-system theory. This topic is particularly relevant because, for many years, dual-system theories have focused on intuitive versus deliberate processes but have not fully addressed how these types of processes coexist and influence each other in situations requiring complex cognition.

The literature on complex cognition and problem-solving, such as the work of Dörner and Funke [40] on complex problem-solving (CPS) and Cronin et al. [41] on dynamic decision-making (DDM), suggests that decision-making in real-world settings involves an intricate blend of cognition, motivation, and emotional regulation. This literature focuses on how people respond to high-stakes, unpredictable environments, where dual-process theories often fail to capture the dynamic and adaptive nature of human decision-making. The literature on complex cognition, e.g., [42,43,44], provides valuable insights into decision-making as a multi-faceted process. This approach illustrates the importance of incorporating complex, interacting variables such as cognitive load, contextual factors, and adaptive strategies that people employ under varying degrees of uncertainty. Recognizing these nuances within the dual-system approach can deepen our understanding of how intuitive and deliberate processes operate jointly to support complex decision-making.

Accordingly, the current paper seeks to expand dual-system theory and discuss it in broader and more nuanced terms. It focuses on dual systems and not the important topic of complex decision-making. I start by reviewing the two classical classifications for the decision-making process models, followed by an overview of the factors that might limit their general acceptability. Then, I present two case studies that use existing results of laboratory studies to illustrate the application of the proposed framework and its potential. Finally, I use novel analyses of existing data to provide initial support for the proposed theoretical assumptions. Thus, while this work primarily synthesizes existing literature on dual-system theories and decision-making, it provides illustrative case studies and initial empirical support to substantiate the presented theoretical perspectives.

## 2. Two Systems of Reasoning

One question in cognitive psychology concerns the issue of whether the mind should be treated as having different functional parts. Simplistically, classic discussions tend to consider two minds at work: one is based on intuitive, automatic processing, and the other is based on reflective, deliberate processing that forms coherent, justifiable sets of beliefs and action plans [45]. While these dual-process models may come in many forms (e.g., heuristic and systematic [46]; experiential and rational [18,47]; intuitive and analytic [48]; reflexive and reflective [49]; and associative and rule-based [20,50]), they all distinguish cognitive operations that are quick and associated with those who are slow and rule-governed [19]. Although many names have been ascribed to these two cognitive mechanisms, the neutral or generic terms System 1 and System 2 proposed by Stanovich and West [21] and adopted by Kahneman and Frederick [27] are used. As in Kahneman and Frederick’s [27] summary, I use the term ‘system’ to describe a collection of cognitive processes that are architecturally (and evolutionarily) distinct and severalized by their speed, controllability, and the contents on which they operate.

System 1. System 1 is based on preconscious, intuitive, and automatic processing. Information is processed rapidly and in parallel; processing is associative, effortless, and opaque to the decision-maker. As such, System 1 places minimal demands on cognitive resources and acts upon schemas that are primarily generalizations from concrete, emotionally significant, intense, or repetitive experiences.

System 2. System 2 is based on slow, deliberate, and reflective information processing in a controlled and self-aware fashion. Information processing is serial, involving deductive reasoning. As such, it is effortful and cognitively demanding. System 2 attains beliefs and knowledge by conscious learning from explicit sources (e.g., books and lectures). Thus, System 1 and System 2 learn from the experience. However, it does so not through automatically established associations but by logical or rational inference. Finally, unlike System 1, System 2 has a short evolutionary history.

The two systems are assumed to operate in parallel, and both processes compete to determine the final responses. In short, when people are required to choose, System 1 processes some of the information (usually the most accessible) and immediately proposes an intuitive answer. In parallel, System 2 monitors the quality of System 1’s response, which may approve, alter, or override. If the overt response retains the initial proposal of System 1 without (much) modification, it is called intuitive. Nonetheless, System 2’s responses will likely remain anchored on initial impressions [18,20,27,48]. The relative contribution of each system is determined by situational factors [51,52] and the decision-maker [21,51,53,54,55,56,57,58,59,60].

The notion of two minds in one brain (i.e., the dual-process model for human cognition) has been empirically confirmed in numerous studies in the past (for a review, see [19,20,26,27,51]). For example, in recent research, Bago et al. [56] asked participants to evaluate a series of fake and real news headlines. First, the participants were asked to respond intuitively (under time limitations and cognitive load). Then, they were asked to rethink their intuitive response without limitations, thus providing a more deliberative response. The results showed that intuitive thinking led to higher rates of believing fake news than deliberative ones. These results might highlight the difference between intuitive and deliberative reasoning in evaluating misinformation. Nevertheless, the structure of the decision-making tasks in this study may have primed participants to question their intuitive responses, potentially leading them toward a more deliberate mode of reasoning. This design choice could affect the natural occurrence of intuitive judgments, as participants might feel compelled to reassess their initial reactions rather than rely on instincts. This bias provides context for interpreting the findings of Bago et al.’s study [56], as the priming effect may lead participants to engage in more deliberate processes than they would in a real-life, non-experimental setting. As such, it may not be sufficient to fully explore complex decisions.

Neuropsychological research [57,58,59] supports the dual-system approach. For example, using fMRI methodology, Goel and colleagues [57,58] showed a neural differentiation of intuitive and deliberate reasoning. Deliberate reasoning was associated with activation of the right inferior prefrontal cortex, whereas intuitive, belief-based responses were associated with activation of the ventral medial prefrontal cortex. In addition, their data supported the idea that System 2 processes can intervene or inhibit System 1 processes. Considering that dual-process models can account for many basic phenomena in psychology in general (e.g., the Müller-Lyer illusion [22] and the moon illusion [60]) and in behavioral decision research in particular (e.g., the ratio bias [61]; the belief–bias effect [26]; and the representativeness, availability, and anchoring heuristics [62]), it is hardly a surprise that the dual-process approach gained attraction, both at the theoretical and applied levels. In Section 4.1, I propose that the predictions of the dual-system approach may be more robust than previously considered in that they can be detected as disequilibrium between behavioral and autonomic physiological responses.

## 3. Compensatory vs. Noncompensatory Models

Another classic theoretical classification of decision-making models is compensatory versus noncompensatory frameworks [2,63,64,65,66]. In general, compensatory models like Expected Utility Theory [67] and Prospect Theory [68,69] assume that differing choices require trade-offs between alternatives and probabilities. For example, when people choose between two gambles (e.g., 100% to obtain USD 100 or 50% to obtain USD 200), they sum and weigh all the available information and choose the alternative with the highest expected utility, at least from their subjective viewpoint. Thus, in a compensatory process, the high features of one alternative (e.g., a big payoff) can compensate for the low levels of others (e.g., low probability). As such, the relationship between the expected values of the choice options is crucial in the decision process.

Conversely, noncompensatory models like Elimination by Aspects [70] and Lexicographic Theory [71] assume that the different features of the options are evaluated in the order of their validity. If the most valid feature (i.e., the “best”) can differentiate between the options (i.e., is clearly better), then the option with the highest value on this feature is chosen, and the decision process is over. Otherwise, the next best feature is examined, and so on. Thus, noncompensatory models assume no tradeoffs between conflicting values of the different features, and low-value features cannot outweigh high-value ones. This means that decisions can be based on limited information (e.g., one feature) while ignoring all others. In addition, Second, noncompensatory processes prevent the need for more complex processes like summing and weighting the possible options [13].

While many studies have confirmed the fact that noncompensatory models, such as fast-and-frugal heuristics, show predictive accuracy under certain conditions, e.g., [72,73,74], there is an ongoing debate as to their felicity as process models, e.g., [10,15,34,75]. Thus, further investigation is required to evaluate whether noncompensatory models merely reflect, to some extent, researchers’ trade-off between simplicity and descriptive accuracy. Tversky [70] pointed out concerning his noncompensatory model (i.e., Elimination by Aspects) that “…there may be many contexts in which it provides a good approximation to much more complicated compensatory models and could thus serve as a useful simplification procedure…” (p. 298). Alternatively, the theoretical appeal of the noncompensatory models might lie not only in their parsimonious nature but also in their ability to describe the cognitive processes that underlie choice behavior.

These questions were addressed by Ayal and Hochman [12]. In their study, two experiments were presented that juxtapose predictions derived from two prototypical fast-and-frugal noncompensatory models (i.e., the Priority Heuristic [13] and Take the Best Heuristic [76]) and alternative predictions derived from compensatory principles. Dependent measures, including reaction time, proportions of the correct response, and level of confidence, were found to be better predicted by compensatory indices; however, these indices could not account for the entire decision process exhaustively. In alignment with these findings, a model based on both types of processes was a runner-up in a behavioral prediction tournament [77]. Thus, these findings highlight the importance of integrating compensatory and noncompensatory principles in choice behavior models aiming to capture the complex decision-making process [40].

## 4. Reconsidering the Dual-System Approach

### 4.1. Simultaneous Contradictory Belief

Sloman [20] defines Criterion S as a decision situation in which people simultaneously feel that two contradicting responses are plausible, even if they do not act upon it [37,78]. In these kinds of situations, “…people first solve a problem in a manner consistent with one form of reasoning and then, either with or without external prompting, realize and admit that a different form of reasoning provides an alternative and more justifiable answer” [20], p. 11.

Many phenomena satisfy Criterion S. Some of the best-known and compelling examples are the Müller-Lyer illusion [22], the conjunction fallacy [79], and superstitious beliefs [80]. However, since people do not affirm both responses, some less apparent phenomena often fail to be recognized as belonging to this category. The effect of this failure is the inability of the field of decision research to explain some contradictory results on some of the most eminent phenomena in the literature (see, for example, Schurr and Erev [81] notions on base-rate neglect; Erev et al. [82], Ert and Erev [83], and Yechiam and Hochman findings on loss aversion [84,85]). Empirical findings regarding physiological measures in general, e.g., [86,87,88,89], and those that combine behavioral and physiological measures in particular, e.g., [84,90,91,92], highlight the potential of this method in identifying such decision phenomena [87], as well as the potential of behavior and autonomic indices to serve as markers of the different cognitive systems [16].

### 4.2. Between-System Processes vs. Within-System Processes

At first glance, the two classifications of the process models as intuitive/deliberate and compensatory/noncompensatory appear to be interdependent. A review of the theoretical and empirical evidence in behavioral decision research suggests that by and large, compensatory processes are considered more rational and deliberate. In contrast, noncompensatory processes are more intuitive.

However, a thorough investigation suggests that this might not be the case. It is not always possible to distinguish between non-/compensatory attributes and their association with Systems 1 and 2. Moreover, findings sometimes suggest that several attributes cannot easily be mapped onto one specific system. For example, the dual-process framework claims that System 2 evolved late as a powerful general-purpose reasoning system. In accordance with this assumption, it has been argued that the effortless, rapid, domain-specific, noncompensatory fast-and-frugal heuristics (e.g., the recognition heuristic [93]) pertain to this system, which is considered to be more deliberate and rational. However, the recognition heuristic draws solely on attributes such as recognition and familiarity, which are considered characteristics of System 1 cf. [27]. Under such assertions, strict classifications of the characteristics of the processes underlying each system might underestimate the possible role of System 2 in the overall decision process.

Considering this situation, I propose that intuitive vs. deliberate characteristics represent between-system processes, whereas compensatory vs. noncompensatory principles represent within-system processes. I believe that any attempt to characterize the cognitive processes that underlie choice behavior under a dual-system approach would benefit greatly from these considerations. To test the robustness and applicability of such a framework, I suggest using indirect measures of processes and combining such indices with physiological ones.

## 5. Dissociation Between Behavior and Autonomic Responses

Within the dual-system view, neuropsychological measures are assumed to provide a clear window into the intuitive system cf. [94]. Moreover, findings show that intuitive reasoning increases the arousal of autonomic indices [90,95]. I argue that the dissociation between behavior and neuropsychological responses can explain contradictory findings in judgment situations under the dual-system framework.

According to prospect theory [68], individuals are more sensitive to losses than equivalent gains. This behavior, which is empirically well-established [96,97,98,99], is considered the manifestation of the basic psychological phenomenon of loss aversion. For example, Tom et al. [99] used functional magnetic resonance imaging (fMRI) to explore brain activity while participants decided whether to accept or reject mixed gambles (i.e., an equal chance to win or lose some money). The authors found that the possibility of losing was associated with decreased activity in brain regions assumed to code subjective values and not increased activity in regions associated with negative emotions. In addition, their results provided evidence that the algebraic function that maps monetary incentives to subjective values is markedly steeper for losses than gains. In addition, loss aversion has been used to account for several paradoxical phenomena in classical decision research, such as the equity premium puzzle [100] and the status quo bias [101].

However, recent evidence shows people do not exhibit loss aversion [82,83,102,103]; for a review, see [24]. Thus, based on Solman’s [20] Criterion S, I postulate that conflicts between the two systems might emerge in situations that examine loss aversion. Accordingly, revisiting these findings using a variety of experience-based tasks and autonomic system measures should yield the expected dissociation between the two systems. In the next sub-sections, I present two case studies to support this claim. The first is a new analysis of the data published in Hochman and Yechiam [84]. I show that examining the difference between pupil diameter and behavioral responses in experience-based risky decisions under the proposed framework can help better understand the unique role of losses in decision-making. In the second, I provide more information on data briefly mentioned in Hochman et al. [16], which used peripheral arterial tone (PAT). Importantly, in both case studies, I do not present novel data. Rather, I try to draw new conclusions from the existing data to illustrate the potential of the proposed theoretical framework.

### 5.1. Case Study 1: Between-System Processes vs. Within-System Processes

Pupillometry measures the extent to which the pupils dilate due to external stimuli or arousal. Previous research found that pupil diameter increases in response to increased processing demands [104,105,106,107,108]. For example, pupil diameter increased during problem-solving (i.e., mental division problems) until the point of the solution, and peak dilations were the largest for the most difficult problems [102]. Problem-solving in real-world contexts involves complex cognitive processes encompassing analytical reasoning and emotional, motivational, and experiential components. For example, in personnel selection [109], decision-makers assess candidates not only on quantifiable skills but also on attributes like adaptability, social fitting, and growth potential. This requires a nuanced understanding of how multiple cognitive processes like intuition, experience-based judgment, and deliberate reasoning converge to inform hiring decisions. Similarly, in political decision-making [110], leaders’ and policymakers’ decisions often extend beyond simple cost–benefit analysis; they involve strategic thinking, empathy, and consideration of long-term societal impacts. This context requires decision-makers to rely on immediate, intuitive judgments and deliberate processes, balancing short-term pressures with broader policy objectives. Recognizing problem-solving as an inherently multifaceted and adaptive process provides a more accurate representation of how individuals navigate complex environments.

Thus, a dissociation between pupil diameter behavioral responses should help clarify contradictory findings in the literature, such as those concerning loss aversion under the dual-system framework. In Hochman and Yechiam’s [84] Study 1, 25 undergraduates were asked to choose between two options repeatedly, each associated with different potential monetary gains or losses. After each selection of one of the options, the obtained payoff in the current trial and an update of the accumulated payoff counter thus far were presented. Participants were instructed to maximize their total earnings. In addition, they were told that in each round, their choice would either lead to a gain or a loss. No further information regarding the payoff structure was provided.

This study had two within-subject conditions, with 60 trials each. In the Mixed Condition, a selection of one of the buttons, referred to as “Risky”, provided a payoff according to the following distribution: 50/50 chance of either gaining or losing two points. The payoff from the alternative button, referred to as “Safe”, was sampled from the distribution with a 50/50 chance of either gaining or losing one point. In the Gains Condition, a fixed value of four points was added to all payoffs to create an all-gains domain. Otherwise, the Gains Condition was identical to the Mixed Condition. Payoffs were delivered deterministically, i.e., each study started with either a gain/relative-gain or a loss/relative-loss and switched to a payoff from the opposite domain in *t* + 1. In addition, on every trial, a constant of 0.1–0.5 points (in 0.1 intervals) was randomly added or subtracted from the sampled payoffs.

Paired-sampled *t*-tests revealed that the aggregated selection in the risky option across all trials was 0.46 in the Mixed Condition and 0.51 in the Gains Condition. These results replicate the ones that were found in Erev et al. [82] and suggest that, behaviorally, participants did not exhibit loss aversion, whether the loss was absolute (i.e., in the Mixed Condition) or relative (i.e., in the Gains Condition). Thus, at least under these conditions, losses did not loom larger than gains, so the expected value function was not steeper in the negative than in the positive domain.

In contrast, absolute losses were associated with higher levels of arousal (indexed by pupil diameter). Moreover, these differences were significant in the epochs of 625 ms to 1125 ms after the stimulus onset, corresponding to previous reports on stimulus recognition, e.g., [111]. Similar findings were not observed in the all-gains domain (i.e., Gains Condition). These findings support the dissociation between behavioral and autonomic measures. Even though the overt response did not reflect loss aversion, the intuitive or automatic response for gains was markedly different than for losses. Thus, it could be argued that loss aversion satisfies Criterion S under certain conditions (e.g., experience-based decisions) since, intuitively, people exhibit higher sensitivity to losses than gains, but their final strategy is similar under the two conditions [24]. This gap can account for previous contradictory results regarding the loss aversion phenomena within the dual-system framework.

Under these assertions, one possible explanation for the dissociation between behavior and autonomic responses could be that in laboratory conditions (in which losses and gains are less real and/or significant), the intuitive tendency to avoid losses is mitigated by other tendencies such as the desire to be perceived as more rational, the tendency to diversify between outcomes, i.e., a diversification bias [112], to increase one’s interest in the task, and so forth. As participants learn that the risk is not big and the alternatives associated with losses are not necessarily detrimental, they learn to override the intuitive response and not shy away from these losses. Further research is needed to examine the plausibility of these interpretations and explore additional ones.

### 5.2. Case Study 2: Peripheral Arterial Tone vs. Behavioral Responses

Peripheral Arterial Tonometry (PAT) is a tool that measures vascular tone at the fingertip and can be used as a non-invasive measure of sympathetic nervous system activity [113]. Vascular tone is influenced by blood pressure, peripheral vascular resistance, blood volume in the finger, and autonomic nervous system activity [114]. Inferred from the signal is the activity of the sympathetic nervous system. The PAT signal (i.e., vasoconstriction) has been found to decrease in response to increased processing demands, e.g., [86]. Thus, much like pupil dilation, the PAT signal has the potential to highlight dissociations between the two different decision systems. Next, I present novel data that used PAT to examine the dissociation between behavioral and physiological responses to losses and gains. A summary of these results was published by [16]. Unlike in the previous case study [84], the data on which the current case study is based was not published in detail previously. Thus, the current case study includes more detailed information about its method.

#### 5.2.1. Method

Design and procedure

Twenty undergraduates were presented with a “money machine” identical to the one described in the previous case study (and detailed in [84]). This study also had two within-subject conditions, with 60 trials each. In the Mixed Condition, a selection of the risky option provided a payoff from one of two distributions. In the gain domain, the possible payoffs were 8.5, 6, and 3.5 points; in the loss domain, the possible payoffs were −1.5, −4, and −6.5. All the payoffs had an equal probability of being sampled. In addition, on each trial, the sampled payoff was drawn from the opposite distribution (i.e., gain or loss) compared to the distribution sampled in trial *t* − 1. Selecting the safe option yielded a constant payoff of one point. In the Gains Condition, a constant payoff of 10 points was added to all payoffs. At the end of this study, participants were compensated based on their selections (around 10 NIS on average).

Because PAT signals are long-lasting waveforms (i.e., several seconds are required to return to base level), the interstimulus intervals (ISIs) were set to 15 s to avoid overlapping responses. Physiological data. Peripheral arterial tone was measured using a finger probe and the SitePAT_200 electrical plethysmograph (Itamar Medical Ltd., Caesarea, Israel).

#### 5.2.2. Results and Brief Discussion

The results of this examination show no behavioral indications for loss aversion in the Mixed Condition. The aggregated P(Risky) across participants was 0.52 (SD = 0.27) and not above chance level (t(19) = 0.393, *p* = 0.699). In contrast, the PAT signal (which represents the average vasoconstriction in a 5-s interval starting from the onset of the stimulus) was significantly different for gains than for losses (t(16) = −1.829, *p* < 0.05 in a one-tailed *t*-test). Although this is a very small sample, these findings replicate previous ones [84] and further support the potential of the dissociation between behavioral and autonomic measures to mark the different decision processes.

### 5.3. Between-System Processes vs. Within-System Processes

To validate the assertion that loss aversion manifests as physiological arousal, which may, under certain conditions, be masked behaviorally, one must demonstrate that in situations that do not satisfy Criterion S, no dissociation between psychological indices and behavioral responses will be found (i.e., the overt response and the physiological arousal both serve as indicators of loss aversion). A study that can create loss-aversive situations in which the overt behavior will correspond to the physiological response, namely, situations in which both the physiological and behavioral responses will exhibit loss aversion, is presented below. In this study, participants will be presented with a set of description-based mixed risky choice decisions of the type:

Choose between
$$\begin{array}{ll} \text{A: Win amount x for sure} & \text{B: Win amount y (probability p)}\\ & \text{Lose amount y (probability 1 − p)} \end{array}$$
in which all the choice problems provide similar expected values.

To induce aversive loss behavior, the potential losses should become more tangible. The payoff structure will be constructed as follows: A fixed payment (about USD 5) will be used as a participation fee. However, the final payment will be a function of the participant’s performance, and no truncation at zero will be included. Thus, non-loss-aversive behavior might result in the possibility of paying the experimenter a relatively substantial amount of money at the end of this study. Of course, this money will eventually be returned to the participants (by participating in an additional study that will provide them with a substantial gain relative to the lost amount). However, this will be disclosed only at the end of the loss-aversion study and after the debtor participant has paid off her bets.

Physiological data will be collected using Pupillometry HR, or PAT. This design creates a situation in which the option that is not associated with the possibility of losing is perceived as more attractive in both systems, as the safer option provides an option to maximize total earnings while avoiding the possibility that might lead to losses. In this case, I would expect no dissociation between physiological and behavioral measures, that is, an increase in arousal in response to the risky options and a decrease in the selection of this option.

## 6. Processes Within the Two Systems

One implication of the notion that intuitive equals noncompensatory and deliberate equals compensatory is that an intuitive decision (which is simple and noncompensatory) is more prone to judgmental errors compared to the more rational, compensatory, and deliberate ones cf. [26,27]. However, this traditional assumption sometimes fails to account for existing empirical evidence. Specifically, in some cases, it has been shown that intuitive and rapid processes might be more accurate than deliberate ones [55,115,116,117]. For example, Glöckner [34] showed that most individuals can integrate multiple pieces of information very quickly and intuitively (with a median decision time of less than three seconds) in a weighted compensatory manner. Similarly, research by Hochman and Erev [118] suggests that decision-makers may intuitively base their preferences on a small sample of previous experiences under similar contingencies. Finally, Ayal et al. [119] show that the quality of the decision depends not on the system but rather on the compatibility between the system and the demands of the task at hand. Tasks that require more deliberate processes benefit from System 2 reasoning, but tasks that require intuitive judgments benefit more from System 1 thinking style.

On the other hand, other research suggests that deliberate and prolonged reasoning, which draws on most (if not all) of the available information in an exhaustive manner, may, under certain conditions, be more prone to judgmental errors. Examples are myopic loss aversion [93], i.e., the tendency to evaluate outcomes frequently; the effect of forgone payoffs [120]; and the Perceived Diversity Heuristic [121]. In all these cases, it has been shown that deliberate reasoning on additional information may lead to a decision that impairs maximization. Moreover, some cases suggest that this less rational behavior might result from deliberate noncompensatory considerations such as initial attraction and (over)generalizing rare outcomes [120].

In the following section, I provide initial support for the plausibility of the claim that the compensatory/noncompensatory classification represents within-system processes and that such a perspective can account for some of the discrepancies in the literature. In their work, Gigerenzer and colleagues [122] introduced a set of noncompensatory process models, like the Priority Heuristic [13] and Take The Best Heuristic [81], for preferences and inferences. As Brandstätter et al. [13] have argued, “The priority heuristic is intended to model both choice and process. It not only predicts the outcome but also specifies the order of priority, a stopping rule, and a decision rule.” (p. 427).

As process models, fast-and-frugal heuristics lend themselves to testable predictions concerning processes. Thus, as Ayal and Hochman [12] have suggested, examining these models vis-à-vis compensatory principles can highlight the nature of the cognitive processes underlying choice behavior.

### 6.1. Examining the Processes Underlying Risky Choices

To test the compensatory against noncompensatory models, the different models must make different predictions on the choices. Thus, to examine the nature of the cognitive processes underlying decisions under risk, I juxtaposed predictions derived from a prototypical noncompensatory model, i.e., the Priority Heuristic [13], and alternative predictions derived from a simple compensatory model (i.e., a simple model of expected value).

Priority Heuristic (PH) is a simple lexicographic model that describes the decision process of people who make preferences. This fast-and-frugal heuristic describes the process of choosing between two alternatives of the type: “the probability p to win amount x, and the probability (1 − p) to win amount y” (X, p; Y, 1 − p). The PH suggests that three hierarchical rules govern this process:

Rule 1: If the difference between the minimum payoffs of the two options exceeds 10% of the maximum payoff (referred to as the aspiration level), select the option with the higher minimum payoff.

Rule 2: If Rule 1 does not apply, examine if the difference between the probabilities of the minimum payoffs exceeds 0.1. If it does, select the option with the lowest probability of obtaining the minimum payoff.

Rule 3: If neither Rules 1 nor 2 apply, choose the option with the higher maximum payoff.

To examine the underlying processes of such choice behaviors, I analyze the response time (RT) and choice proportion (CP) in a replication of the choice problems used by Brandstätter et al. [13] to provide an empirical examination of the PH as a process model. The data are taken from Ayal and Hochman [12], but the analyses are novel. As suggested by Brandstätter et al. [13], I classified the sets into a 2 (one reason or three reasons examined) × 2 (gambles of similar or dissimilar expected value) mixed-factorial design. This classification enables a formulation of contradictory hypotheses for RT and CP, one that fits the PH and the other to compensatory principles.

Reaction time (RT). Noncompensatory models assume a sequential, limited search for information with a clear stopping rule: the search ends when the decision-maker finds a piece of information that distinguishes between options. Thus, noncompensatory models predict that people will require a higher reaction time with more information (e.g., cues, reasons) they examine. By contrast, compensatory models suggest that the decision-maker integrates all the relevant information. As a result, compensatory models predict that when more information integration is required or when the integration between arguments leads to smaller differences between the alternatives (e.g., a small difference between the alternatives’ expected values), response time should be longer. Thus, I can make the following hypotheses:

**Hypothesis** **1.**
*RT_PH_: Reaction time will be higher in the three-reason-examined choice problems than in the one-reason-examined choice problems.*


**Hypothesis** **2.**
*RT_EV_: Reaction time will be higher in the similar-expected-value choice problems relative to the dissimilar-expected-value choice problems.*


Choice proportion (CP). In preference tasks, choice proportion describes the proportion of choices decision-makers make that aligns with the prediction of a specific choice model. Assuming that people can make processing errors on each step of their decision strategy, it can be argued that the earlier the examination is terminated, the fewer errors will be involved in the final decision. Therefore, noncompensatory models predict that when the examination is terminated after the first argument, the resulting choice proportion should be more in line with the strategy’s predictions (e.g., 80/20) than when termination occurs after the second reason (e.g., 70/30) cf. [13]. Alternatively, if people use compensatory strategies, the choice proportion should be highest for decisions that are derived more easily (e.g., when the different arguments are better at distinguishing between the choices). Thus, I can make the following hypotheses:

**Hypothesis** **3.**
*CP_PH_: The proportion of choices aligned with the PH will be higher in the one-reason-examined choice problems than in the three-reason-examined choice problems.*


**Hypothesis** **4.**
*CP_EV_: The proportion of choices aligned with the EV will be higher in the dissimilar-expected value choice problems than in the similar-expected value choice problems.*


In the study of Ayal and Hochman [12], 50 undergraduates were instructed to make 20 choices between gambles of the sort $X, p; $Y, 1 − p. All possible outcomes were gains. The mean RT for the one-reason-examined choice problems was 12.31 s (SD = 8.34) when the two options had similar EVs, and 10.03 s (SD = 5.40) when the two options had dissimilar EVs. Similarly, the mean RT for the three-reasons-examined choice problems with similar EVs was 10.9 s (SD = 6.18) and 8.78 s (SD = 4.34) for the dissimilar EV choices. Repeated measures analysis of variance (ANOVA) revealed a significant main effect both for the level of similarity between the EVs (F(1, 47) = 11.158, *p* < 0.001) and the number of reasons examined (F(1, 47) = 4.178, *p* < 0.05). Importantly, the results of the number of reasons examined contradict the prediction of the PH. Specifically, the results demonstrate that it takes longer for decision-makers to choose between alternatives that require one piece of information (according to the noncompensatory model) than three pieces, results that support Hypothesis RT_EV_.

A similar pattern was observed for CP. The mean CP (i.e., the choice proportion in line with the model’s predictions) was 0.55 (SD = 0.22) when the two options had similar EVs and 0.85 (SD = 0.19) when they had dissimilar EVs. Likewise, for the three-reasons-examined choice problems, the mean CP was 0.6 (SD = 0.23) for similar EVs and 0.892 (SD = 0.18) for dissimilar EVs problems. Repeated measures ANOVA revealed a significant main effect for the level of similarity between the expected values (F(1, 47) = 119.913, *p* < 0.001). However, there was no main effect for the number of reasons examined (F(1, 47) = 2.708, ns), nor was there a significant interaction between the two variables. Thus, again, we see support for the compensatory and not the noncompensatory model.

In summary, the analyses of Ayal and Hochman’s [12] data suggest that despite the value of noncompensatory principles, it does not capture the full extent and complexity of decision-making processes. The pattern of results obtained from different measures (i.e., RT and CP) supports the idea that when making preferences, people tend not to rely on limited information in a noncompensatory manner but rather to integrate all available information. Nevertheless, CP did not reduce to chance level in the dissimilar expected value choice problems (one-sampled *t*-test revealed that the aggregated mean CP across reasons examined was 0.57, t(47) = 3.476, *p* < 0.001). This could suggest that when integrating information does not help differentiate between the two options, people use additional strategies (either compensatory or not) to make their choice.

In this context, Brandstätter et al. [13] acknowledged these conclusions and admitted that their model could be better at predicting preferences if it assumed that people compute the EV of each option, take their ratio into account, and choose the option with the highest EV only if the ratio exceeds 2. The current results suggest that Brandstätter et al.’s intuition was correct. The coexistence of compensatory and noncompensatory principles may lead to better decision-making models [65] that help better understand complex decision-making processes [40].

### 6.2. Further Modeling Analysis

Although the existing data highlight the importance of accounting for both compensatory and noncompensatory principles, a more direct examination of the specific cognitive processes underlying intuitive and deliberate reasoning in these situations is in order. Notwithstanding, I present an alternative process model based on the two systems’ view as compensatory/noncompensatory and the previously reported results. According to this (tentative) model, decision-makers begin their decision-making process with a preliminary intuitive compensatory process that integrates all or most available information. For example, this integration is used to evaluate the expected values of the available options. The decision is made if this initial compensatory process provides clear evidence toward a specific option (e.g., points to an option with a substantially larger EV). If this is not the case and the initial process does not differentiate between the options, decision-makers use a rational compensatory selection among one of the many noncompensatory tools available at their disposal (i.e., heuristics and rules of thumb) to reach the best decision while investing minimal time and effort.

To provide initial support for the proposed model, I applied a new analysis to the data of Ayal and Hochman [12]. Since intuitive reasoning is rapid (i.e., short), if it is a compensatory process, I can assume that in ‘easy’ choice problems (i.e., with dissimilar expected values), all short decision times will lead to a higher proportion of maximization (i.e., higher CP). This is because if the maximizing alternative is identified intuitively, any additional reasoning might add more noise to the decision, resulting in greater judgmental errors. On the other hand, if the intuitive mechanism for these kinds of choice problems is noncompensatory, I would expect it to be less efficient as the number of reasons examined increases (due to cumulative error). To examine these assertions, the results of the RT only for the ten dissimilar EV choice problems were classified into a 2 (one reason or three reasons examined) × 2 (short or long response time) array. This enables us to make the following hypotheses:

**Hypothesis** **5.**
*INTUITIVE_PH_: The proportion of choices aligned with the PH will be higher in the one-reason-examined choice problems than in the three-reason-examined choice problems, regardless of the response time.*


**Hypothesis** **6.**
*INTUITIVE_EV_: The proportion of choices aligned with the EV will be higher when the response time is short than when the response time is long.*


#### Summary of the Results and Brief Discussion

Across participants, the mean CP for the one-reason-examined choice problems was 0.87 (SD = 0.35) when the response time was short (i.e., less than 5 s, M = 3, Md = 3.5), and it decreased to 0.81 (SD = 0.37) when the response time was long (i.e., more than 10 s, M = 16.6, Md = 14.4). For the three-reason-examined choice problems, the mean P(maximizing) was 1.00 (SD = 0.00) for short response times and 0.75 (SD = 0.38) for long response times. A repeated-measures ANOVA was conducted to test the effects of the number of reasons examined and the response time on choice behavior. The ANOVA included 2X2 within-participant independent variables with P(maximizing) as the dependent variable. This analysis revealed a significant main effect for the response time (F(1,7) = 3.723, *p* < 0.05 in a one-tail test). No main effect for the number of reasons examined (F(1, 7) = 0.179, *p* = 0.67) was found, nor was there a significant interaction between the two variables (F(1, 7) = 0.396, *p* = 0.55).

The results of the current analysis support Hypothesis INTUITIVE_EV_. Namely, the finding that judgment accuracy was higher for extremely rapid responses replicates previous results [34] and suggests that the complex compensatory integration of outcomes and probabilities was employed intuitively.

As mentioned, the current analysis is just an initial attempt to examine the proposed model. To examine the generality of these results, researchers need to collect additional data on description-based choice decisions. In addition, the predictions of the proposed model should be juxtaposed with other process models, such as the Decision Field Theory (DFT) [123]. The DFT assumes that at each moment in time, the decision-maker intuitively thinks about the various payoffs of each prospect and produces an affective reaction (i.e., valence) to each prospect accordingly. These valences are integrated across time to produce the preference state at each moment. A threshold controls the stopping rule for this process: the first prospect to reach the top threshold is chosen.

According to DFT [123], higher thresholds necessitate reaching a stronger state of preference, which allows decision-makers to obtain more information about the possible options. This extends the deliberation process and enhances accuracy. In contrast, lower thresholds permit decisions based on weaker preference states, limiting the information acquisition, thereby shortening the deliberation process and reducing accuracy (i.e., a tradeoff between speed and accuracy). The threshold is assumed to be low under high and high under low time pressure. In contrast, the proposed model suggests the opposite. Since intuitive reasoning is considered compensatory, the model assumes that decreasing accuracy results from acquiring additional information and a prolonged noncompensatory reasoning process.

To test our model against the DFT [123], one might design a study where participants face description-based decisions under risk. This study will include a 2 (time limitation and the magnitude of the dissimilarity between the alternative’s expected values) × 3 (low, medium, and high) factorial within-subject design. While the DFT predicts a reversal of opinion and decreasing accuracy as time constraint increases, the proposed model predicts the opposite. Namely, the model assumes that rational reasoning was obtained very early. Thus, any additional computation time will only result in adding noise to the system. This noise is predicted to impair reasoning.

Manipulating the magnitude of the dissimilarity between the expected values is offered to ensure that under the low time constraint condition, participants will still be forced to rely on deliberate reasoning. In addition, physiological data can be collected to examine dissociations between intuitive and deliberate processes.

## 7. Summary

The current review highlights the importance of a nuanced understanding of dual-system models, emphasizing the critical distinction between and within intuitive and deliberative processes. While the current review did not deal with complex decision models directly and focuses on the dual-system approach, it suggests that integrating complex cognition with dual-system theory can enrich its explanatory power by addressing real-world complexities often absent in controlled settings. Traditionally, dual-system models treat intuitive (System 1) and deliberate (System 2) processes as distinct and relatively fixed in their roles. However, frameworks like CPS [40] and DDM [41] reveal that intuitive and deliberate processes can adapt based on the complexity and immediacy of the decision context. For example, intuitive judgments in complex environments are not merely “quick and dirty” solutions. Rather, they are shaped by previous experiences and learned patterns, while deliberate reasoning may adaptively simplify to meet time or other environmental constraints.

This alignment suggests that intuitive and deliberate processes may not solely belong to distinct systems but could function within overlapping, flexible systems influenced by context, experience, and cognitive demands. Recognizing this adaptive capacity within dual-system models enhances our theoretical comprehension. This differentiation enriches our theoretical comprehension of cognitive mechanisms and significantly enhances decision-making models’ predictive capability. By utilizing the interplay between compensatory and noncompensatory processes within each system, the novel analyses reported here resonate with but diverge from existing literature with the potential to provide some interesting insight. As such, the current review might suggest that while some decision-making models adequately capture certain dimensions of cognitive processing, they often fall short of encapsulating the full spectrum of human complexity. If this is the case, it seems that judgment and decision-making scholars would benefit from a shift towards more granular analyses within cognitive psychology and decision research.

The current review may also point to other limitations of applying the dual-system approach to real-world scenarios, particularly regarding ecological and external validity. It might be argued that the structured, simplified dilemmas typically used in laboratory settings do not accurately reflect the complexity of decisions encountered in daily life, where factors like time constraints, emotional stakes, and social influences have real consequences [124,125]. Real-world decision-making often involves layered and dynamic considerations that are difficult to capture with compensatory or non-compensatory models alone, as these models generally assume a fixed set of criteria and outcomes [65]. Furthermore, ecological validity is compromised when decision-making scenarios lack contextual relevance, leading to choices that may not generalize beyond the lab [126]. Researchers have emphasized the need for decision models that account for adaptive, heuristic processes that align more closely with individuals’ intuitive responses in everyday settings [127,128]. By acknowledging these limitations, the dual-system framework can be viewed as a foundational yet partial representation of decision-making that requires further adaptation to better capture the complexities of real-life choices.

Of course, several limitations, like potential biases in the literature selection and interpretation, might hinder these conclusions. Moreover, empirical investigations with diverse methodologies across several contexts are crucial to validating and refining the proposed theoretical approach. Finally, integrating theoretical frameworks from decision-making theory, behavioral economics, psychology, and physiology presents unique challenges, particularly when bridging methodologies from the natural sciences and economic theory. Behavioral economics, for instance, offers abstract constructs (e.g., risk aversion and loss aversion), which serve as simplified representations of complex human behavior in economic contexts [68]. These constructs were designed to model reality at a high level, and as such, verifying or refuting them directly may appear to oversimplify nuanced psychological processes. However, examining physiological responses and behaviors to economic situations can provide a more comprehensive view of decision-making by revealing underlying cognitive and emotional mechanisms not always captured in economic models alone [129]. Incorporating insights from psychology and physiology offers a unique opportunity to explore how foundational economic principles manifest in real-time decision processes. While methodologically complex, this interdisciplinary approach seeks to enrich our understanding of decision-making by examining the interactions between economic rationality and physiological responses, adding depth to the traditionally separate fields of economic and psychological research [130]. By acknowledging these methodological distinctions, I wish to offer a multidisciplinary approach to leverage the strengths of each field while carefully navigating the limits of cross-disciplinary assumptions. Thus, this line of research could yield some important theoretical and practical implications, from improving behavioral interventions to crafting more effective marketing strategies.

## Data Availability

No new data were created for this article.

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
