# Peer review of "Beyond the Surface: A New Perspective on Dual-System Theories in Decision-Making"

_behavsci, 2024, doi:10.3390/bs14111028_

Round 1

Reviewer 1 Report

Comments and Suggestions for Authors

Thank you for giving me the opportunity to provide feedback on the manuscript entitled “Beyond the Surface: A New Perspective on Dual-System Theories in Decision-Making,” which was submitted for consideration to Behavioral Sciences. This paper offers a unique evaluation of the dual-system approach, challenging the traditional classification that links intuitive processes exclusively with non-compensatory models and deliberate processes with compensatory models. 

Overall, my evaluation of this paper is somewhat favorable, but some negative aspects require further attention. The topic is relevant, and the paper's objectives are valuable. It is worthwhile for researchers in decision sciences to work towards a more refined and comprehensive theoretical account of the dual-system framework. Gaining a better understanding of the interplay between these two systems and their respective roles in reasoning merits our attention. The author has tried to simplify complicated concepts, and I appreciated their ability to connect different theoretical models. Additionally, the author has undertaken an interesting exercise by attempting to link the latest research on the dual system approach to using peripheral psychophysiological measurement tools and techniques.

The paper, as it stands, is not yet ready for publication. In my opinion, the manuscript still lacks clarity. I believe it requires a comprehensive review and more interdisciplinary integration. The following comments are based solely on my expertise and concentrate on the relevant elements within my research interests. I have refrained from commenting on concepts or potential angles I am unfamiliar with. Here are my suggestions for improving the paper:

1.       If the author aims to provide a “critical evaluation” of the dual system approach, they must discuss research on complex cognition. I believe this is the main weakness of the paper, as it explores the dual-system analogy in depth but gives little attention to the idea that it remains an analogy of more complex cognitive systems. Several authors have pointed out this fact in recent years. The central blind spot here is the lack of any mention of studies from or related to the naturalistic paradigm (e.g., Klein, 2008), which is dedicated to studying how people make decisions in cognitively demanding situations that resemble the real world. In my view, one cannot aspire to critique the dual process approach without referencing the abundant literature on complex decision-making. At the very least, two traditions should be included and discussed in the “Introduction” section: the European perspective, such as the complex problem-solving (CPS) framework (e.g., Dörner & Funke, 2017), and the literature (mostly American) relating to dynamic decision-making (DDM) (e.g., Cronin, Gonzalez, & Sterman, 2009). The author should mention these because they have studied human decision-making as a complex process for several years, focusing on the interplay between cognition, motivation, and emotions (Funke, 2010). The author himself alludes to this point in the summary section (lines 587-590): “As such, the current review might suggest that while some decision-making models adequately capture certain dimensions of cognitive processing, they often fall short of encapsulating the full spectrum of human complexity.”

 2.      The author also introduces the concept of problem-solving (line 245) as a mental division problem, which, in my opinion, is somewhat reductive. I think the author must underline that problem-solving is in itself decision-making and that it entails a diverse array of complex cognitive processes. Several examples can illustrate that decision-making processes manifest complex cognition. As the author mentions, these mechanisms are better studied through a combination of behavioral measurement tools such as eye tracking, laboratory-simulated decision-making tasks, and psychophysiological instruments. In order to clearly illustrate this body of knowledge, I recommend that the author explores and refer to recent efforts made by various researchers in studying complex decision-making in interdisciplinary fields such as personnel selection (Landers & Sanchez, 2022), political decision-making (Béchard, Hodgetts, Morneau-Guérin, Ouimet, & Tremblay, 2023), and education (Herde, Wüstenberg, & Greiff, 2016). The author does not need to reorganize the central claim made by the paper; however, mentioning such studies is important for aligning the paper with a perspective that goes 'beyond the surface' of the current state of the literature on dual-process theory.

 3.       In its current form, the manuscript needs clarification regarding the nature of the study. In the 'Summary' section on line 581, the author referred to it as a ‘review,’ but it seems to be an empirical study of two use cases. The author should specify the type of study from the outset and avoid confusion by labeling it as a review. In this regard, the method section needs to be expanded to provide details of the empirical protocol proposed by the study. This includes the study's research design (e.g., laboratory experiment, quasi-experimental design?) and the procedure (e.g., the settings in which the 'money machine' was presented to participants), participant characteristics, recruitment process, sampling, and recruitment period.

 4.       In line with the previous point, a brief paragraph discussing the potential limitations of the dual system approach to describe how decision-making processes happen in the real world would also be welcome (i.e., ecological validity, external validity). Some authors raise this point, noting that the type of dilemmas studied with compensatory/non-compensatory models are rarely, if ever, encountered in real life. This would help contextualize the author's approach and highlight its nature.

Here is a short list of details that need to be revised to ensure the fluidity of the paper:

· The introduction seems too abrupt and somewhat lacks fluidity. For example, the author refers to their ‘manuscript’ in the body of the text (line 21). The section on indirect measures (lines 45-50) is a good example. It would be beneficial to explain the difference between indirect and direct measures from the outset, as this is not common knowledge, and these measures define the paper's focus.

· This is a minor detail, but it must be thoroughly addressed. On line 242: “An eye-tracking apparatus measures the extent to which the pupils dilate due to external stimuli or arousal.” In my opinion, this statement is incorrect. Eye tracking primarily monitors and measures eye movements and gaze patterns to determine where a person is looking, how long they are looking at specific areas, and how their eyes move between different points (saccades, fixations). Pupillometry, on the other hand, measures changes in pupil dilation. From the outset, it would be more accurate to state that pupillometry is […].

· The measure in the study by Bago (referred to on page 3) may be questioned because the task itself primes individuals to question their intuition as erroneous. Perhaps this should be mentioned by the author.

· Line 294: “Case Study 2 […]” - What is Case Study 1 then? This should be included in the title of the section.

Comments on the Quality of English Language

There are some typographical errors (e.g., 'underling,' line 26; 'Solman,' line 232; 'makeing,' line 408, etc.). The manuscript requires a thorough revision of the English.

Author Response

First, I wish to thank the reviewer for this thorough review and for seeing value in my work and helping me improve it. 

Comment 1: If the author aims to provide a “critical evaluation” of the dual system approach, they must discuss research on complex cognition. I believe this is the main weakness of the paper, as it explores the dual-system analogy in depth but gives little attention to the idea that it remains an analogy of more complex cognitive systems. Several authors have pointed out this fact in recent years. The central blind spot here is the lack of any mention of studies from or related to the naturalistic paradigm (e.g., Klein, 2008), which is dedicated to studying how people make decisions in cognitively demanding situations that resemble the real world. In my view, one cannot aspire to critique the dual process approach without referencing the abundant literature on complex decision-making. At the very least, two traditions should be included and discussed in the “Introduction” section: the European perspective, such as the complex problem-solving (CPS) framework (e.g., Dörner & Funke, 2017), and the literature (mostly American) relating to dynamic decision-making (DDM) (e.g., Cronin, Gonzalez, & Sterman, 2009). The author should mention these because they have studied human decision-making as a complex process for several years, focusing on the interplay between cognition, motivation, and emotions (Funke, 2010). The author himself alludes to this point in the summary section (lines 587-590): “As such, the current review might suggest that while some decision-making models adequately capture certain dimensions of cognitive processing, they often fall short of encapsulating the full spectrum of human complexity.”

Response: I completely agree with the reviewer. While the current work aims to address the issue of two systems directly, it was a complete oversight from my side not to relate to the important topic of complex decisions. While the focus of the work is still the dual system approach, I now acknowledge this important topic and relate to it and its connection to my work in the revised introduction and summary section, as well as in the manuscript itself. I also thank the reviewer for referring me to relevant literature, which is now cited and mentioned in the text.

Comment 2: The author also introduces the concept of problem-solving (line 245) as a mental division problem, which, in my opinion, is somewhat reductive. I think the author must underline that problem-solving is in itself decision-making and that it entails a diverse array of complex cognitive processes. Several examples can illustrate that decision-making processes manifest complex cognition. As the author mentions, these mechanisms are better studied through a combination of behavioral measurement tools such as eye tracking, laboratory-simulated decision-making tasks, and psychophysiological instruments. In order to clearly illustrate this body of knowledge, I recommend that the author explores and refer to recent efforts made by various researchers in studying complex decision-making in interdisciplinary fields such as personnel selection (Landers & Sanchez, 2022), political decision-making (Béchard, Hodgetts, Morneau-Guérin, Ouimet, & Tremblay, 2023), and education (Herde, Wüstenberg, & Greiff, 2016). The author does not need to reorganize the central claim made by the paper; however, mentioning such studies is important for aligning the paper with a perspective that goes 'beyond the surface' of the current state of the literature on dual-process theory.

Response: I completely agree. Thank you for this important observation. I now address this important point in the revised version and better explain how it relates to my work.

Comment 3: In its current form, the manuscript needs clarification regarding the nature of the study. In the 'Summary' section on line 581, the author referred to it as a ‘review,’ but it seems to be an empirical study of two use cases. The author should specify the type of study from the outset and avoid confusion by labeling it as a review. In this regard, the method section needs to be expanded to provide details of the empirical protocol proposed by the study. This includes the study's research design (e.g., laboratory experiment, quasi-experimental design?) and the procedure (e.g., the settings in which the 'money machine' was presented to participants), participant characteristics, recruitment process, sampling, and recruitment period.

Response: I apologize for the lack of clarity. I agree that the nature of the paper was not well explained, and I thank the reviewer for pointing me to this important issue. The introduction was extensively revised, and I made the goals and methods clearer both in the intro and in the description of the methods themselves.

Comment 4: In line with the previous point, a brief paragraph discussing the potential limitations of the dual system approach to describe how decision-making processes happen in the real world would also be welcome (i.e., ecological validity, external validity). Some authors raise this point, noting that the type of dilemmas studied with compensatory/non-compensatory models are rarely, if ever, encountered in real life. This would help contextualize the author's approach and highlight its nature.

Response: The proposed paragraph was added to the revised version of the summary.

Comment 5: The introduction seems too abrupt and somewhat lacks fluidity. For example, the author refers to their ‘manuscript’ in the body of the text (line 21). The section on indirect measures (lines 45-50) is a good example. It would be beneficial to explain the difference between indirect and direct measures from the outset, as this is not common knowledge, and these measures define the paper's focus.

Response: The introduction was extensively revised based on the reviewer's comments.

Comment 6: This is a minor detail, but it must be thoroughly addressed. On line 242: “An eye-tracking apparatus measures the extent to which the pupils dilate due to external stimuli or arousal.” In my opinion, this statement is incorrect. Eye tracking primarily monitors and measures eye movements and gaze patterns to determine where a person is looking, how long they are looking at specific areas, and how their eyes move between different points (saccades, fixations). Pupillometry, on the other hand, measures changes in pupil dilation. From the outset, it would be more accurate to state that pupillometry is […].

Response: I completely agree. The text was changed accordingly.

Comment 7: The measure in the study by Bago (referred to on page 3) may be questioned because the task itself primes individuals to question their intuition as erroneous. Perhaps this should be mentioned by the author.

Response: This point was added to the revised text.

Comment 8: · Line 294: “Case Study 2 […]” - What is Case Study 1 then? This should be included in the title of the section.

Response: I am sorry for this typo. The words "Case study 1" were the title of section 5.1. I also cleared the idea and essence of the case studies and their meaning.

Comment 9: There are some typographical errors (e.g., 'underling,' line 26; 'Solman,' line 232; 'makeing,' line 408, etc.). The manuscript requires a thorough revision of the English.

Response: I am sorry for the typos. The paper underwent a thorough English editing. I hope all typos were corrected.  

Reviewer 2 Report

Comments and Suggestions for Authors

The work undoubtedly deserves attention. The approaches proposed by the authors to the analysis of hypotheses and assumptions of the theory of decision-making and behavioral economics, based on the tools of psychological research, can arouse keen interest among specialists in the relevant subject areas, as well as among simply interested readers.

As comments and wishes to the author (or authors), I would like to draw attention to the following points.

(1)           Methods involving the conjugation of heterogeneous theoretical concepts (decision-making theory, behavioral economics on the one hand, psychology and physiology on the other) are questionable. Including from the position of their adequate scientific seriousness. The basic assumptions of behavioral economics, such as risk aversion, the assumption that losses are more important than gains, etc., are essentially fundamental abstract theoretical constructs, initially formulated as some simplification (model representation) of reality. Therefore, attempts to confirm or refute them may look somewhat naive. For example, no serious researcher is currently attempting to construct a real or realistic utility function. This, however, does not invalidate classical microeconomic theory. Thus, the existence of a certain methodological dissonance between natural-scientific and economic-theoretical approaches cannot be denied. Perhaps the authors should provide more detailed explanations on this issue.

(2)         Overall, the presentation is explicitly verbal. It contains econometric fragments. However, research papers adopt a somewhat different style of presenting the results of econometric analysis. I assume that the current appearance of the paper is due to the stylistic traditions of a particular scientific school.

(3)           I have not been able to form correct ideas about the volumes and representativeness of the samples on the basis of which the authors draw their conclusions. Perhaps this information should be emphasized more clearly. The paper would have benefited from a clearer justification of the choice of statistical methods.

(4)           The part 7 “Summary” of the paper seems overly general and vague.

(5)           In some places, there is first-person narration. For example: “Considering this situation, I propose that ....” (line 206) Is this style acceptable?

(6)      An error in line 49. It is written [26, 287]. Probably should be [26-28].

Comments on the Quality of English Language

I am not an expert in English language stylistics, but I believe there are possibilities for improvement in the language of the paper.

Author Response

First, I wish to express my gratitude to the reviewer for taking the time to review the manuscript and for the important and helpful comments. Without a doubt, they helped me improve my work.

Comment 1: Methods involving the conjugation of heterogeneous theoretical concepts (decision-making theory, behavioral economics on the one hand, psychology and physiology on the other) are questionable. Including from the position of their adequate scientific seriousness. The basic assumptions of behavioral economics, such as risk aversion, the assumption that losses are more important than gains, etc., are essentially fundamental abstract theoretical constructs, initially formulated as some simplification (model representation) of reality. Therefore, attempts to confirm or refute them may look somewhat naive. For example, no serious researcher is currently attempting to construct a real or realistic utility function. This, however, does not invalidate classical microeconomic theory. Thus, the existence of a certain methodological dissonance between natural-scientific and economic-theoretical approaches cannot be denied. Perhaps the authors should provide more detailed explanations on this issue.

Response: Thank you for this important comment. In the revised version, I address this issue in length as one of the possible limitations of my work (in the summary section).

Comment 2: Overall, the presentation is explicitly verbal. It contains econometric fragments. However, research papers adopt a somewhat different style of presenting the results of econometric analysis. I assume that the current appearance of the paper is due to the stylistic traditions of a particular scientific school.

Response: I am sorry it wasn't clear enough that the paper is mainly a review work with minimal empirical demonstrations. Thus, the style is not traditional. However, in line with your (and the other reviewer's) comments, I completely revised the introduction and better explained the aim and method of the manuscript. The entire paper was revised accordingly.

Comment 3: I have not been able to form correct ideas about the volumes and representativeness of the samples on the basis of which the authors draw their conclusions. Perhaps this information should be emphasized more clearly. The paper would have benefited from a clearer justification of the choice of statistical methods.

Response: Thank you for this comment. I provide more details about the research I review and better explain its significance. Importantly, the current work includes no novel data, and this is why it is not an empirical work. I am sorry it wasn't clear enough in the previous version, and I thank you for pointing out this important issue. 

Comment 4: The part 7 “Summary” of the paper seems overly general and vague.

Response: I agree. Thank you very much for encouraging me to revise and improve the summary of the paper extensively.

Comment 5:   In some places, there is first-person narration. For example: “Considering this situation, I propose that ....” (line 206) Is this style acceptable?

Response: I understand it might seem odd. It is more common in my field (JDM). However, I am happy to change to narration if it is inappropriate for Behavioral Sciences. 

Comment 6: An error in line 49. It is written [26, 287]. Probably should be [26-28].

Response. Thank you. The typo was corrected (in should be [26, 28].

Comment 7: I am not an expert in English language stylistics, but I believe there are possibilities for improvement in the language of the paper.

Response: Thank you. I made significant changes to improve the English and readability of the paper.

Round 2

Reviewer 1 Report

Comments and Suggestions for Authors

Thank you to the author for their careful revision of the manuscript. The new version more effectively frames the dual-process theory as an analogy for more complex cognitive processes. Additionally, the author incorporates relevant literature on complex problem-solving.

The revised manuscript effectively integrates the naturalistic paradigm and enhances the dual-process approach by exploring complex decision-making. It examines the European perspective on complex problem-solving (CPS) and dynamic decision-making (DDM), strengthening its theoretical foundation.

The author references relevant scientific literature to highlight the research efforts of various scholars in different fields who aim to understand decision-making as a complex interplay of intuitive and deliberate processes. It is now easier to understand that dual-process theory is relevant in "simplifying" this complexity, enabling a systematic study of the complex cognitive processes involved in decision-making.

The nature of the study is now clearer, especially in the introduction and summary sections. The distinction between the two case studies is now clearer. Additionally, the summary section is better structured, helping the reader understand the potential and scope of dual-process theory while also emphasizing the model's limitations. In this regard, the author’s effort to broaden the scope of the discussion is noteworthy.

Comments on the Quality of English Language

There are still a few typos in the manuscript that should be corrected before publication.

Reviewer 2 Report

Comments and Suggestions for Authors

I won't claim that the changes made have radically changed the article. However, I have no principal objections to its publication.

Comments on the Quality of English Language

From my point of view, the language of the article is quite acceptable for understanding its meaning.